# Ustekinumab Promotes Radiological Fistula Healing in Perianal Fistulizing Crohn’s Disease: A Retrospective Real-World Analysis

**DOI:** 10.3390/jcm12030939

**Published:** 2023-01-25

**Authors:** Jiayin Yao, Heng Zhang, Tao Su, Xiang Peng, Junzhang Zhao, Tao Liu, Wei Wang, Pinjin Hu, Min Zhi, Min Zhang

**Affiliations:** 1Department of Gastroenterology, Guangdong Provincial Key Laboratory of Colorectal and Pelvic Floor Disease, The Sixth Affiliated Hospital, Sun Yat-Sen University, Guangzhou 510655, China; 2Department of Colorectal Surgery, Guangdong Provincial Key Laboratory of Colorectal and Pelvic Floor Disease, The Sixth Affiliated Hospital, Sun Yat-Sen University, Guangzhou 510655, China

**Keywords:** Crohn’s disease, ustekinumab, perianal fistula, radiological fistula remission

## Abstract

There is insufficient evidence to confirm the efficacy of ustekinumab (UST) in promoting fistula closure in perianal fistulizing Crohn’s disease (CD) patients. We aimed to evaluate the efficacy of UST in a real-world setting. The data were retrospectively analyzed. Intestinal clinical and endoscopic changes were evaluated. Fistula radiological outcomes were determined using the Van Assche score. A total of 108 patients were included, 43.5% of whom had complex perianal fistulas. Intestinal clinical and endoscopic remission was achieved in 65.7% and 31.5% of patients, respectively. The fistula clinical remission and response rates were 40.7% and 63.0%, respectively, with a significant reduction in Perianal Crohn’s disease Activity Index [5.0(3.0, 8.0) vs. 7.5(5.0, 10.0), *p* < 0.001] and Crohn’s Anal Fistula Quality of Life [23.5(9.3, 38.8) vs. 49.0(32.3, 60.0), *p* < 0.001]. Radiological healing, partial response, no change, and deterioration were observed in 44.8%, 31.4%, 13.4%, and 10.4% of patients, respectively. The cut-off UST trough concentration for predicting fistula clinical remission was 2.11 μg/mL with an area under the curve of 0.795, a sensitivity of 93.3%, and a specificity of 67.6%. UST is efficacious in promoting radiological fistula closure in patients with perianal fistulizing CD. A UST trough concentration over 2.11 μg/mL was correlated with a higher likelihood of perianal fistula clinical remission.

## 1. Introduction

Perianal fistula is the most common complication of Crohn’s disease (CD), affecting approximately 40% of patients [1]. It represents an aggressive phenotype of CD, which is likely to respond poorly to multiple medications, has a high risk of relapse and disease-associated disability, and faces early-onset surgery [2,3]. Patients with perianal fistulizing CD suffer from anal pain, purulent discharge, restricted sexual activity, and abdominal symptoms, which undoubtedly result in a lower quality of life. Therefore, management and monitoring of perianal fistulizing CD remains challenging. 

A multidisciplinary approach is recommended for the treatment of perianal fistulizing CD because of its complexity [4]. According to the global consensus established by Gecse in 2014 [5], for patients with perianal abscesses and active draining fistula, seton or fistulotomy should be performed, followed by aggressive medical therapies. Monoclonal antibodies against tumor necrosis factor (anti-TNF) agents, including infliximab and adalimumab, are effective in perianal fistulizing CD, as shown by the results of the ACCENT II [6] and CHARM [7] trials. However, it should not be ignored that a proportion of patients are primary non-responders to anti-TNF agents, and some have to switch to other biologics targeting different inflammatory pathways due to loss of response or development of severe adverse effects. 

Ustekinumab (UST), an antibody targeting the p40 subunit shared by interleukin 12 and 23, effectively induces disease remission, as supported by the UNITI-1 and UNITI-2 clinical trials [8,9]. Our recently published study demonstrated that clinical and endoscopic remission rates were 84.2% and 73.7%, respectively, at week 16/20 after UST initiation, which adds evidence to the effectiveness of UST in refractory CD [10]. However, there is still no strong evidence supporting the efficacy of UST in treating perianal fistulizing CD, despite a series of post hoc or subgroup analyses [11,12]. 

We aimed to assess the short-term efficacy of UST in treating perianal fistulizing CD, especially in promoting radiological fistula healing, and to evaluate the UST trough concentration for predicting clinical fistula remission. 

## 2. Methods

### 2.1. Study Design

This was a retrospective cohort study based on the data of patients with perianal fistulizing CD from 1 March 2020 to 31 October 2022 at the Sixth Affiliated Hospital of Sun Yat-Sen University (Guangzhou, China). This study was approved by the Ethics Committee of Sun Yat-Sen University (2021ZSLYEC-066) and the Clinical Trial Registry (NCT04923100). Consent from the patients was waived because all the data we used were anonymous. All procedures were performed in accordance with the principles of the Declaration of Helsinki.

### 2.2. Patients

Consecutive patients meeting the following inclusion criteria were included: First, patients underwent comprehensive screening and diagnosis for CD according to internationally accepted criteria [13,14] with supportive clinical, endoscopic, radiological, and histopathological findings. Second, active perianal fistula was confirmed by clinical symptoms and baseline magnetic resonance imaging (MRI). Third, patients were administered UST therapy and followed up until the third infusion at weeks 16 or 20, with a drug interval of q8w or q12w, respectively. Patients with incomplete data, development of severe adverse events, and discontinuation of UST therapy within 16 weeks were excluded.

All patients were first infused with intravenous UST (260 mg for those weighing <55 kg, 520 mg for those weighing >85 kg, and 390 mg for those weighing between 55–85 kg) and subcutaneous UST (90 mg every 8 or 12 weeks) afterward [15]. Perianal surgeries were performed if needed before the initiation of UST infusion. The indications for surgery include the following: (1) acute abscess formation; (2) marked purulent external orifice, which worsens the quality of life; (3) an active fistula revealed by MRI scan with the characteristics including lesion range larger than 1 cm, deep ramification, or multiple ramification formation. The protocols for prior surgery include abscess incision, partial extra-sphinteric fistulotomy or fistulectomy, and loose seton drainage. Concomitant oral antibiotics including metronidazole and ciprofloxacin were prescribed for 4 weeks after surgery. As for the patients with loose seton, a second definite surgical repair or seton removal was evaluated at week 16/20 after UST initiation. The UST trough concentration and antidrug antibodies were detected before the third infusion of UST. Data on patient characteristics, serologic biomarkers (including C-reactive protein [CRP], erythrocyte sedimentation rate, platelets, hemoglobin, and albumin, and imaging were extracted from hospital digital records. 

### 2.3. Definition

CD was classified using the widely accepted Montreal classification system [16]. Crohn’s disease activity index (CDAI) [17], perianal Crohn’s disease activity index (PDAI) [18], and Crohn’s anal fistula quality of life (CAF-QoL) [19] were evaluated at baseline and at week 16/20. Intestinal clinical remission was defined as a CDAI < 150, and intestinal clinical response was defined as a >70 reduction in CDAI and/or CDAI < 150 [17]. Fistula clinical remission was defined as the absence of any draining fistula, and fistula clinical response was defined as a decrease of >50% in the number of draining fistulas according to the fistula drainage assessment index (FDA) [1]. Rutgeerts [20] scores and simple endoscopic score for Crohn’s disease (SES-CD) [21] were used to evaluate the changes in endoscopic findings in patients with or without colectomy, respectively. Endoscopic remission was defined as a Rutgeerts score ≤i1 or SES-CD ≤2 [20,21]. Endoscopic response was defined as a reduction of one grade from baseline in Rutgeerts score or a reduction of >50% in SES-CD [20,21]. C-reactive protein (CRP) normalization was defined as a CRP level of <4 mg/L.

MRI was performed to evaluate the fistula status. The number of fistulas, anatomicalclassification, hyperintensity on the fat-saturated T2 sequence, and track thickness and volume were recorded. A simple fistula was defined as a superficial/inter-sphincteric/trans-sphincteric fistula with only one track, without extension or abscess. Complex fistulas were defined as inter-sphincteric/trans-sphincteric fistulas with more than one track, or supra-sphincteric/extra-sphincteric/rectovaginal fistula [1]. Four MRI-based radiological outcomes were described, including healing, improvement, no change, and deterioration. Radiological fistula healing was defined as the absence of a high-signal track on fat saturated T2 sequences. Improvement was defined as a reduction in the number and volume of fistula, and >10% decrease in the MRI signal. No change was defined as the same in the number of fistulas and the volume of inflammation. Deterioration was defined as an increase in the size and number of fistula tracks [22]. Van Assche scores [23] ranging from 0 to 22 reflected fistula activity, including fistula number, location, extension, hyperintensity on T2, collections, and rectal wall involvement. Two specialists from the Colorectal Department (HZ and BH) diagnosed perianal fistulizing CD and assessed the improvement of perianal fistula based on gentle compression, examination under anesthesia, and MRI scans. Two experienced radiologists (WTC and WRL) read the MRI scans, evaluated the radiological outcomes, and recorded the Van Assche scores. Clinical, endoscopic, and radiological evaluations were recommended at week 16/20.

### 2.4. Statistical Analysis

Continuous data were presented as mean ± standard error (S.D.E) or median with interquartile range (IQR), while categorical data were presented as percentages. Student’s *t*-test or Wilcoxon test was performed to compare indicators before and after UST treatment. A receiver operating characteristic (ROC) curve was established to figure out the cut-off value of UST trough concentration for predicting clinical fistula remission with the area under the curve (AUC), sensitivity, and specificity calculated. All analyses were conducted using SPSS 22.0. A statistically significant *p*-value was defined as a two-sided *p*-value < 0.05. 

## 3. Results

### 3.1. Patients’ Characteristics 

A total of 308 patients diagnosed with CD and receiving scheduled UST treatment were enrolled. Of these, 137 patients were excluded due to the absence of perianal fistula based on clinical symptoms and MRI scans, 51 for insufficient follow-up duration, and 12 for incomplete data (Figure 1). A total of 108 eligible patients were finally included, 74.1% of whom were male, with a mean age of 29.2 ± 1.0 years at diagnosis and a mean disease duration of 4.3 ± 0.4 years. As for the Montreal classification, 61.1% of the patients were assigned to B1 (non-stricturing, non-penetrating) and 71.3% to L3 (ileocolonic) phenotypes. Most fistulas were inter-sphincteric (63.9%), followed by superficial (18.5%), trans-sphincteric (15.7%), and supra-sphincteric (1.9%). Of the fistulas, 43.5% were complex fistulas, with a median baseline Van Assche score of 9.0 (7.0,14.0), as determined by MRI scans. Of the patients, 29.6% had perianal abscesses and 57.4% had proctitis. Among them, 14 patients underwent fistulotomy before UST therapy, 2 of whom received additional ileostomy due to the severe proximal intestinal lesion. The baseline characteristics are listed in Table 1.

### 3.2. Efficacy of UST on CD

After administration of UST, the patients showed less inflammatory burden manifested by a significant decrease in CRP (14.6 ± 2.4 vs. 24.0 ± 3.2, *p* = 0.002), and improved nutrition manifested by an increase in hemoglobin (129.8 ± 2.1 vs. 119.0 ± 2.1, *p* < 0.001) and Alb (40.2 ± 5.7 vs. 36.9 ± 5.1, *p* < 0.001) (Table 2). Intestinal clinical remission was observed in 65.7% of patients, and intestinal clinical response was observed in 71.3% of patients (Figure 2A). CRP normalization was achieved in 55.6% of patients (Figure 2B). A total of 99 patients had endoscopy reexamination, of whom 22 patients were evaluated by Rutgeerts score and 77 by SES-CD. Endoscopic remission and response were achieved in 31.5% and 45.4% of patients, respectively (Figure 2C).

### 3.3. Efficacy of UST on Perianal Fistulas

For all enrolled patients, a marked reduction in PDAI (5.0(3.0, 8.0) vs. 7.5(5.0, 10.0), *p* < 0.001) and CAF-QoL (23.5(9.3, 38.8) vs. 49.0(32.3, 60.0), *p* < 0.001) indicated the mitigation of fistulas (Table 2). Fistula clinical remission was observed in 40.7% and fistula clinical response in 63.0% of patients (Figure 3A). All the patients were required to return at week 16/20 after UST initiation for clinical, endoscopic, and radiological reevaluation. However, a proportion of patients refused MRI reexamination due to disappearance of perianal symptoms, economic burden, or time constraint. Eventually, 62.0% (67/108) of the patients underwent MRI scans. The percentages of patients with fistula healing, partial response, no change, and deterioration were 44.8%, 31.4%, 13.4%, and 10.4%, respectively (Figure 3B). After UST treatment, the Van Assche score significantly decreased (5.5(0.0, 10.0) vs. 9.0(7.0, 14.0), *p* < 0.001), indicating the confirmed amelioration in fistula radiological outcomes (Figure 3C).

### 3.4. Efficacy of UST on Anti-TNF Naïve and Exposure Patients

We further evaluated the efficacy of UST on patients who were anti-TNF naïve and those who had anti-TNF exposure. Intestinal clinical remission rate in anti-TNF naïve patients was significantly higher than that in anti-TNF exposure patients (78.9% vs. 58.6%, *p* = 0.033). There was no significant difference in intestinal clinical response, fistula clinical remission and response, endoscopic remission and response, and radiological remission between the two groups (Table 3). Nevertheless, we did observe more favorable remission and response rates in clinical, endoscopic, and radiological evaluations in anti-TNF naïve patients, although not statistically significant, according to the subgroup analysis.

### 3.5. Relationship of CD Clinical Remission and Clinical Fistula Response

Fistula clinical fistula remission/response was observed in 80.3% of the patients who had achieved intestinal clinical remission, but only 43.2% in those who did not, indicating that intestinal clinical remission positively correlated with fistula clinical remission.

### 3.6. Exposure–Response Effect of UST on Perianal Fistulizing CD

Overall, 64 patients had UST trough concentration detected at week 16/20 after initiation of UST. The median UST trough concentration at week 16/20 was 2.4 (0.9, 3.5) μg/mL. In a quartile analysis of UST trough concentrations, we demonstrated that fistula clinical remission and response rates correlated with UST trough levels. Higher rates of fistula clinical remission and response were observed in the higher UST trough concentration group (Figure 4A). A significantly higher fistula remission and response rate was found in the higher UST trough concentration quartile. The cut-off UST trough concentration predicting clinical fistula remission was 2.11 μg/mL, with an AUC of 0.795, a sensitivity of 93.3%, and a specificity of 67.6% (Figure 4B). Figure 5 showed a typical case manifesting radiological fistula healing after UST therapy in patients with perianal fistulizing CD.

## 4. Discussion

In this study, approximately 40% of the patients achieved fistula clinical remission after UST initiation. Of note, 44.8% of the patients achieved deep radiological fistula healing according to post-treatment MRI. Our clinical and radiological results verified the acceptable short-term efficacy of UST for perianal fistulizing CD, particularly in promoting radiological fistula closure. 

Infliximab was the first proven effective biologic in promoting and maintaining CD-related fistula closure, supported by high-quality randomized controlled trials (RCTs) with fistula closure as the primary endpoint [24]. According to a multicenter, double-blind RCT conducted by Daniel et al. [6], 40% of patients had a complete fistula response at week 54 after scheduled infliximab administration. Adalimumab is effective in treating fistulizing CD, however only with low-grade evidence [25]. Majority of studies reported that 30–50% of patients achieved clinical fistula remission after long-term anti-TNF therapies [6,26,27]. Our results manifested that 40.7% of the patients presented fistula closure after initiation of UST, which was similar to those reported previously. Given that this study focused solely on the short-term efficacy of UST, favorable long-term outcomes may be expected.

UST is the second-line biologic recommended for perianal fistulizing CD. A post hoc analysis of UNITI-1/UNITI-2 reported that 24.7% of patients achieved fistula closure at week 8 and 80% of patients achieved clinical fistula response at week 44 after UST treatment [28]. The BioLAP study [29], including 207 patients with perianal CD, was a retrospective trial with the largest sample size reported to date. Therapeutic success was achieved in 38.5% of patients treated with UST. A prospective observational study in the Netherlands reported that 35.7% (10/28) of patients achieved clinical fistula remission 24 weeks after UST initiation [30]. However, RCTs are still lacking with regards to fistula closure as the primary endpoint to evaluate the efficacy of UST on perianal fistula. 

UST was first approved for the treatment of CD in 2016 in America and in 2020 in China. The efficacy of UST on CD has rarely been reported in China, and has never been reported in perianal fistulizing CD. To our knowledge, this is the first real-world study conducted in Chinese population to report the effectiveness of UST in perianal fistulizing CD. The fistula clinical remission rate was 40.7%, similar to that reported previously [31]. We focused more on radiological outcome regarding that radiological fistula healing always lags behind fistula clinical remission and calls for greater efforts to realize it. However, our results showed approximately 45% of patients had achieved radiological fistula healing, which undoubtedly adds our confidence in efficacy of UST in the treatment of perianal fistulizing CD. Moreover, a better performance of UST was seen on patients who were biologic-naïve manifested by the significantly higher rate in intestinal clinical remission according to our subgroup analysis, which was consistent with the well-known SUCCESS trial [32]. Nevertheless, there was no significant difference in the effectiveness of UST in promoting clinical or radiological fistula healing in biologic-naïve and biologic-exposure patients.

Pelvic MRI is a pivotal tool for perianal fistula diagnosis, classification, severity evaluation, and monitoring. Radiological fistula healing continues after clinical fistula closure, for internal tracks may persist despite closure of the external opening, leading to a higher rate of relapse [33]. Patients who achieve radiological fistula remission may maintain fistula resolution, regardless of continuation or discontinuation of anti-TNF therapy [34]. In this study, all eligible patients had a precise diagnosis and classification of perianal fistula based on MRI scans. In addition, 62.0% (67/108) of the patients underwent MRI scans at the post-therapy follow-up. The radiological fistula healing rate was 44.8%, indicating the ideal efficacy of UST for complete fistula closure. Follow-up imaging can assist with disease monitoring and therapeutic management. 

Perianal fistulizing CD exerts profound effects on patient’s psychosocial state and daily life [35]. To date, limited data have been obtained regarding the effect of perianal fistulas on quality of life. The PDAI is widely used to measure CD-associated perianal disease activity. It is neither specific to perianal fistulas nor patient-centered [36]. CAF-QoL is the first disease-specific and patient-reported outcome index in clinical practice, involving factors such as burden of symptoms and treatment, and negative impact on quality of life [19]. In this study, we combined the PDAI and CAF-QoL to evaluate the impact of perianal fistulas on patients with CD. Favorable changes in both the PDAI and CAF-QoL were observed after UST therapy. 

It has been reported that an IFX concentration of 12 μg/mL is associated with fistula remission. Optimizing biologics correlates with a higher response rate in perianal fistulizing CD patients [37,38]. Nevertheless, no studies have yet proposed a cut-off UST trough level associated with fistula healing. Sands et al. concluded that perianal fistula resolution is not associated with a higher UST serum concentration [39]. In contrast, one observational study noted that 50% of patients with UST escalation into q4w or q6w administration intervals achieved a clinical response in perianal disease [40]. In this study, we did manifest the exposure–effect relationship between clinical fistula remission and UST trough levels. The cut-off value of UST we reported was 2.11 μg/mL, which was much higher than 1.12 μg/mL, the cut-off value of UST associated with clinical remission (defined as CDAI < 150) that we reported previously [10]. Undoubtedly, more high-quality studies are needed to further verify the relationship between UST escalation and fistula outcome. 

This study had some limitations. First, this was a single-center study with a relatively small sample size. The evidence from this retrospective study should be validated further in a larger sample size from multiple IBD centers nationwide. Moreover, we only manifested the short-term efficacy of UST in perianal fistulizing CD with short-term follow-up; hence, and the long-term efficacy and the safety of UST for perianal fistula was not evaluated. The strengths of this study include strict definitions, radiological evaluation combined with clinical assessment, and emphasis on quality of life. 

## 5. Conclusions

In conclusion, UST is effective in promoting clinical and radiological fistula remission in patients with CD. A trough concentration of UST higher than 2.11 μg/mL was associated with clinical fistula remission at week 16/20. More RCTs with fistula closure as the primary outcome are warranted to evaluate the efficacy of UST in treating perianal fistulizing CD in-depth.

## Figures and Tables

**Figure 1 jcm-12-00939-f001:**
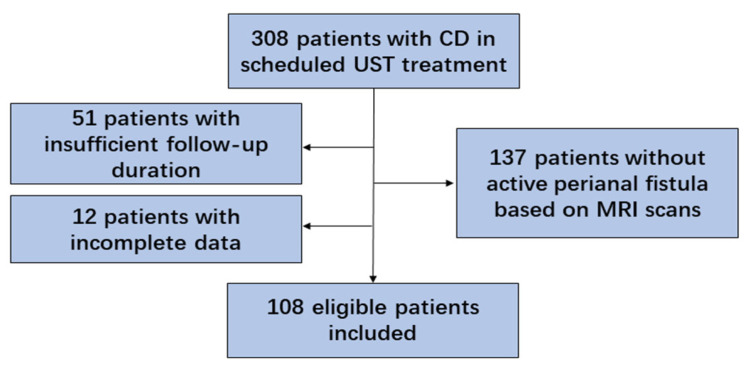
Flow chart of the study (CD, Crohn’s disease; UST, ustekinumab; MRI, magnetic resonance imaging).

**Figure 2 jcm-12-00939-f002:**
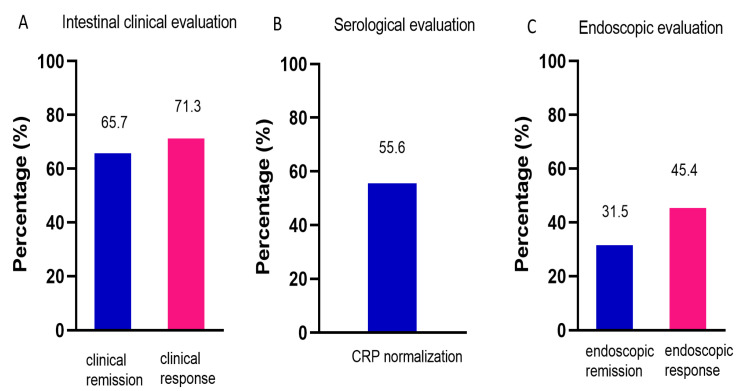
Efficacy of UST on CD. (**A**) Intestinal clinical evaluation using CDAI at week 16/20, *n* = 108. (**B**) Serological evaluation determined by CRP levels, *n* = 108. (**C**) Endoscopic evaluation using Rutgeerts score or SES-CD, *n* = 99. UST: ustekinumab; CDAI: Crohn’s disease activity index; CRP: C-reactive protein; SES-CD: simple endoscopic score for Crohn’s disease.

**Figure 3 jcm-12-00939-f003:**
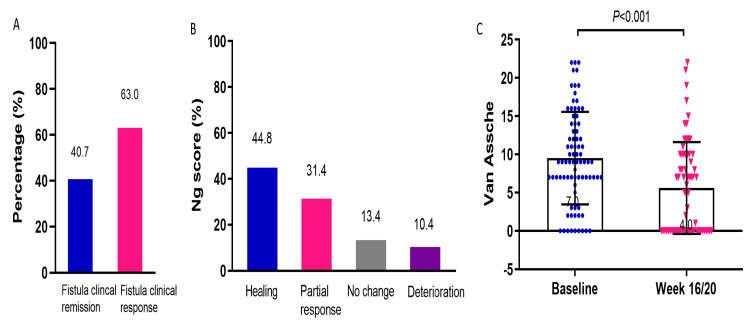
Efficacy of UST on perianal fistula. (**A**) Fistula clinical remission and response rates determined by PDAI, *n* = 108. (**B**) Radiological outcomes evaluated by Ng score, *n* = 67. (**C**) Changes in Van Assche scores before and after UST therapy, *n* = 67. UST: ustekinumab; PDAI: perianal Crohn’s disease activity index.

**Figure 4 jcm-12-00939-f004:**
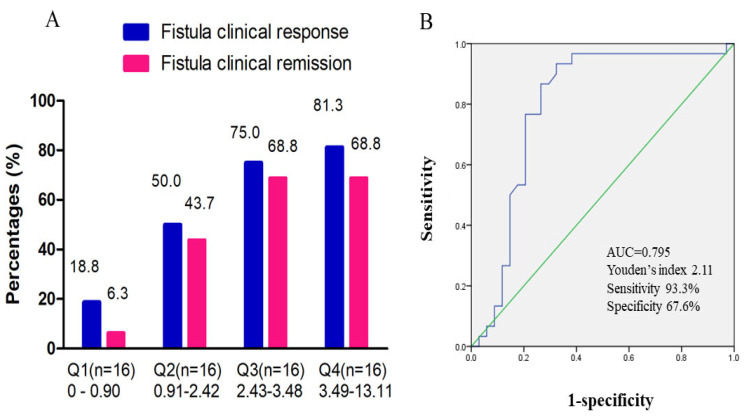
Exposure–response effect of UST on fistula clinical outcome. (**A**) Quartile analysis of UST trough concentration associated with fistula clinical remission and response. (**B**) ROC curve of UST trough concentration at week 16/20 predicting clinical fistula remission. The cut-off UST trough concentration was 2.11 μg/mL, with an AUC of 0.795 [95%CI: 0.675–0.915], a sensitivity of 93.3%, and a specificity of 67.6%. ROC, receiver operating characteristic; UST, ustekinumab; AUC, area under the curve; CI, confidence interval.

**Figure 5 jcm-12-00939-f005:**
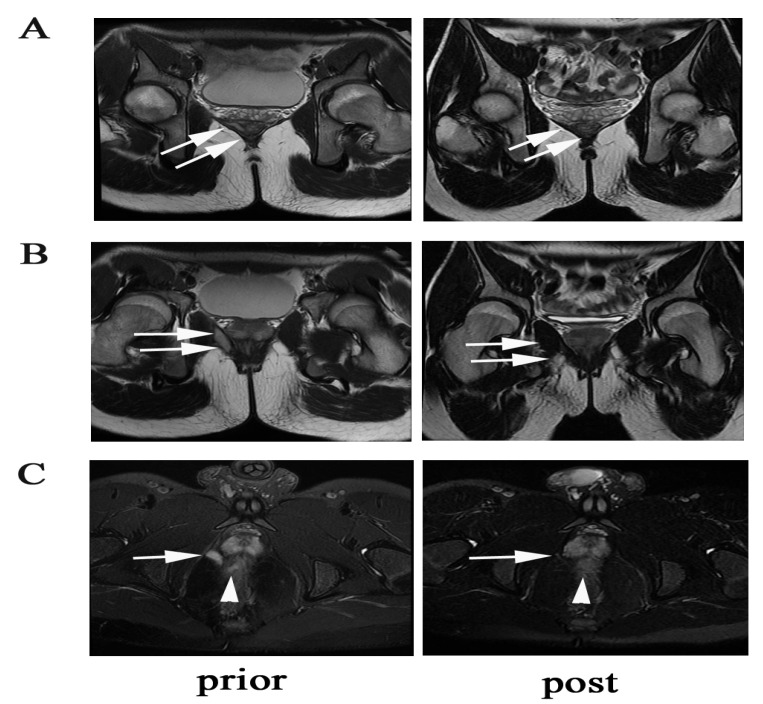
A case showing radiological fistula healing of a supralevator fistula. (**A**) The arrows show the supralevator part of the fistula closure with scarring. (**B**) The arrows show the infralevator part of the fistulas closure with scarring. (**C**) The arrows show vanishment of the supralevator lesion, and the triangle shows improvement of proctitis.

**Table 1 jcm-12-00939-t001:** Baseline characteristics of overall patients.

Variables	Total Patients (*n* = 108)
Male, *n* (%)	80 (74.1)
Age at diagnosis, [years, mean ± S.D.E]	29.2 ± 1.0
Disease duration, [years, mean ± S.D.E]	4.3 ± 0.4
Montreal classification	
Age, *n* (%)	
A1 (≤16 years)	8 (7.4)
A2 (17–40 years)	87 (80.6)
A3 (>40 years)	13 (12.0)
Disease behavior, *n* (%)	
B1 (non-stricturing, non-penetrating)	66 (61.1)
B2 (stricturing)	13 (12.0)
B3 (penetrating)	29 (26.9)
Disease location, *n* (%)	
L1 (ileal)	20 (18.5)
L2 (colonic)	11 (10.2)
L3 (ileocolonic)	77 (71.3)
L4 (upper GI)	21 (19.4)
Fistula type, *n* (%)	
Simple	61 (56.5)
Complex	47 (43.5)
Fistula location, *n* (%)	
Superficial	20 (18.5)
Inter-sphincteric	69 (63.9)
Trans-sphincteric	17 (15.7)
Supra-sphincteric	2 (1.9)
Extra-sphincteric	0 (0)
Van Assche at baseline, median (IQR)	9.0 (7.0,14.0)
Proctitis, *n* (%)	62 (57.4)
Perianal abscess, *n* (%)	32 (29.6)
Previous medication, *n* (%)	
Steroids	50 (46.3)
Immunosuppressants ^1^	76 (70.4)
Anti-TNF agents ^2^	70 (64.8)
Previous intestinal surgery, *n* (%)	27 (25.0)
Extraintestinal manifestation, *n* (%)	8 (7.4)

^1^ Immunosuppressants includes thiopurines, methotrexate, cyclophosphane, and thalidomide. ^2^ Anti-TNF agents refers to infliximab or/and adalimumab. IQR, interquartile range; S.D.E, standard error; GI, gastrointestinal.

**Table 2 jcm-12-00939-t002:** Efficacy of UST on patients with perianal fistulizing CD (*n* = 108).

Variables	Baseline	Week 16/20	*p* Value
Inflammatory burden (mean ± S.D.E)			
CRP (mg/L)	24.0 ± 3.2	14.6 ± 2.4	0.002
ESR (mm/h)	22.8 ± 2.3	18.2 ± 1.6	0.051
Platelet (×109/L)	311.9 ± 9.4	296.4 ± 9.1	0.090
Nutritional state (mean ± S.D.E)			
Hemoglobin (g/L)	119.0 ± 2.1	129.8 ± 2.1	<0.001
Alb (g/L)	36.9 ± 5.1	40.2 ± 5.7	<0.001
BMI	19.0 ± 2.9	19.3 ± 3.3	0.247
Intestinal clinical evaluation (IQR)			
CDAI	179.5 (117.6, 258.2)	112.2 (71.9, 171.8)	<0.001
Fistula clinical evaluation (IQR)			
PDAI	7.5 (5.0, 10.0)	5.0 (3.0, 8.0)	<0.001
CAF-QoL	49.0 (32.3, 60.0)	23.5 (9.3, 38.8)	<0.001

CRP: c-reactive protein; ESR: erythrocyte sedimentation rate; Alb: albumin; BMI: body mass index; CDAI: Crohn’s disease activity index; PDAI: perianal Crohn’s disease activity index; CAF-QoL: Crohn’s anal fistula quality of life; CD, Crohn’s disease; UST, ustekinumab; IQR, interquartile range; S.D.E, standard error.

**Table 3 jcm-12-00939-t003:** Efficacy of UST on patients with anti-TNF exposure and anti-TNF naïve.

Variables	Anti-TNF Naïve ^1^	Anti-TNF Exposure	*p* Value
Intestinal clinical remission, *n*/*n* (%) (*n* = 108)	30/38 (78.9)	41/70 (58.6)	0.033
Intestinal clinical response, *n*/*n* (%) (*n* = 108)	30/38 (78.9)	47/70 (67.1)	0.195
Fistula clinical remission, *n*/*n* (%) (*n* = 108)	20/38 (52.6)	24/70 (34.3)	0.064
Fistula clinical response, *n*/*n* (%) (*n* = 108)	25/38 (65.8)	42/70 (60.0)	0.554
Endoscopic remission, *n*/*n* (%) (*n* = 99)	14/35 (40.0)	17/64 (26.6)	0.381
Endoscopic response, *n*/*n* (%) (*n* = 99)	20/35 (57.1)	25/64 (39.1)	0.260
Radiological remission, *n*/*n* (%) (*n* = 67)	14/25 (56.0)	16/42 (38.1)	0.154

^1^ Anti-TNF agents refers to infliximab or/and adalimumab.

## Data Availability

The data that support the findings of this study are available from the corresponding author upon reasonable request.

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
