# Peer review of "Ustekinumab Promotes Radiological Fistula Healing in Perianal Fistulizing Crohn’s Disease: A Retrospective Real-World Analysis"

_jcm, 2023, doi:10.3390/jcm12030939_

Round 1

Reviewer 1 Report

This is a non-randomized observational study on 108 patients on the efficacy of treatment with ustekinumab for perineal fistulizing Crohn's disease.

The authors evaluated the efficacy of the treatment in relation to the clinical response and inflammatory index on the intestinal disease and on the clinical response and in a part of the patients by means of MR in perianal disease.

They describe a complete remission of perianal disease in 40.7% of cases and a partial response in 63% at 16 or 20 weeks.

There are some aspects that need to be clarified

Most fistulas are simple or intersphincteric, therefore at low risk of complexity and generally with better prognosis.

No data on surgical treatment is reported, in which patients a surgical treatment was performed, according to which criteria and which type of intervention. In most centers, therapy with biological drugs is preceded by surgical exploration under anesthesia to optimization of drainage.

Only in 67 patients underwent control MR, it is not specified in the article what are the selection criteria for doing the MR and if the diagnosis of remission always has been confirmed with MR

Author Response

Response to Reviewer 1:

(1) Our baseline characteristic data showed that 56.5% of the patients had simple fistulas, and 63.9% of fistulas were inter-sphincteric. We agreed with the reviewer. A relatively baseline simple status of fistula may lead to a better prognosis. Therefore, we listed the baseline information as detailed as possible.

(2) Thank you so much for your advices. We can’t agree with you more. We have mentioned that perianal surgeries were performed if needed before the initiation of UST infusion in the 2nd paragraph in the Method-Patients Section. The description was too simple, which we have added detailed surgical information including the surgical criteria and intervention type (2rd paragraph in the Methods-Patients Section). Besides, we have added data on surgical treatment in the Result-Patients’ characteristic Section. Thank you again for your suggestions, which undoubtedly help improve the manuscript.

(3) All the patients were required to return at week 16/20 after UST initiation for clinical, endoscopic, and radiological re-evaluation. However, a proportion of patients refused to undergo pelvic MRI reexamination due to disappearance of perianal symptoms, economic burden, or time constraint. Thank you for your advice. We have added detailed descriptions in the Result-Efficacy of UST on perianal fistula Section to make it more comprehensively understandable (3rd paragraph in the Result Section).

(4) As we have mentioned in the Method-Definition Section that fistula clinical remission was confirmed according to the fistula drainage assessment index (FDA, detailed information could be found in reference 1), and radiological fistula remission was confirmed by MRI (2nd paragraph in the Methods-Definition Section). Thank you so much for your question.

Reviewer 2 Report

In this retrospective cohort study based on the data of patients with perianal fistulizing CD, the Authors aimed to evaluate the efficacy of UST in a real-world setting.

In the presented study, a total of 108 patients were included, 43.5% of whom had complex perianal fistulas. The fistula clinical remission and response rates were 40.7% and 63.0%, respectively. Radiological healing, partial response, no change, and deterioration were observed in 44.8%, 31.4%, 13.4%, and 10.4% of patients, respectively. The cut-off UST trough concentration for predicting fistula clinical remission was 2.11 μg/ml. 

This study will provide valuable information in our knowledge.  However, I do have some suggestions.

Previous Anti-TNF agent usage was 70 (64.8%) in this study. Data regarding prior biologic use, detailed prior anti-TNF therapy failure information was not available in the majority of UST therapy studies for perianal fistulizing CD. If the authors compare the response and remission rates of UST therapy in anti-TNF naive and anti-TNF experienced/failure groups in pCD patients, the data will all provide us with crucial information.

Author Response

Response to reviewer 2:

We can’t agree more with the advice from reviewer. We have added a subgroup analysis to evaluate the efficacy of UST on patients who were biologic naïve and those who had biologic exposure. Notably, intestinal clinical remission was achieved in 78.9% of biologic-naïve patients, which was significantly higher than that of biologic-exposure patients. What’s more, we observed higher rates in endoscopic and radiological remission and response, without statistical significance though, which indicated a better performance of UST on biologic-naïve patients, as we have further discussed in the discussion section. We have added a paragraph (4th paragraph) in the result section, revised the 4th paragraph in the discussion section and added Table 3 and a new reference. Thank you for your advices.

PS: Here are something more to be clarified to the editors:

This manuscript had been sent to Editage for language polishing again before we uploaded the revised version to ensure the accuracy of the language. We have uploaded the language certification as attachment. We had an abstract published in GUT (DOI: 10.1136/gutjnl-2022-IDDF.135) and a preprint (DOI: https://doi.org/10.21203/rs.3.rs-1354312/v1) in 2022 with the same research aim but a really small sample size. In order to draw a precise conclusion to this hot topic, we have enlarged the sample size in this manuscript, which was not published or under consideration (in whole or in part) for publication elsewhere. 

Round 2

Reviewer 1 Report

I have no other comments